# Fresh-Cut Bell Peppers in Modified Atmosphere Packaging: Improving Shelf Life to Answer Food Security Concerns

**DOI:** 10.3390/molecules25102323

**Published:** 2020-05-16

**Authors:** Carla Barbosa, Thelma B. Machado, Manuel Rui Alves, Maria Beatriz P. P. Oliveira

**Affiliations:** 1CISAS-IPVC/Centre for Research and Development in Agrifood Systems and Sustainability-Polytechnic Institute of Viana do Castelo, Av. do Atlântico, s/n, 4900-908 Viana do Castelo, Portugal; cbarbosa@estg.ipvc.pt (C.B.); mruialves@estg.ipvc.pt (M.R.A.); 2REQUIMTE/Dep. Chemical Science, Faculty of Pharmacy, University of Porto, Rua Jorge Viterbo Ferreira, 228, 4050-313 Porto, Portugal; thelma_machado@id.uff.br; 3Postgraduate Program in Sciences Applied to Health Products, Faculty of Pharmacy, Fluminense Federal University. Rua Dr Mário Viana, 523, Santa Rosa–Niterói–RJ 24241-000, Brazil; 4Postgraduate Program in Biosystems Engineering, School of Engineering, Fluminense Federal University. Rua Passo da Pátria, 156 bloco D, sala 236, São Domingos-Niterói-RJ 24210-240, Brazil

**Keywords:** *Capsicum annuun* L., sensory analysis, bioactive compounds, principal component analysis

## Abstract

The influence of modified atmosphere packaging (MAP, 10% O_2_ and 45% CO_2_) on the quality characteristics of fresh-cut green, red and yellow bell peppers (*Capsicum annuum* L. var *annuum*) was investigated. Packaging film bags (Krehalon MLF40-PA/PE) with fresh-cut bell peppers were stored for up to 17 days at 5 °C. The in-package O_2_ level ranged between 10 and 15%, respecting the current recommendations for fresh-cut vegetable products. Initial CO_2_ levels were higher than commonly used (from 5 to 10%), decreasing progressively over time due to the permeability of the selected polyethylene film. At the end of the storage period, they stabilized between 2 and 5%. A small variation in texture, moisture, titratable acidity, pH and microbial growth was observed during the storage period, as well as a good color retention and sensory properties maintenance. Negligible losses in the antioxidant activity and bioactive compounds (total phenol, flavonoid, anthocyanin and carotenoid content) were noted at the end of the study. Sensory analysis showed that panelists could not detect significant differences among sampling periods. A PCA with predictive biplots confirmed the existence of significant correlations. The products retain their initial characteristics without severe loss of quality until at least the 17th storage day. Given the current commercial shelf life of fresh-cut bell peppers, ranging from 9 to 14 days, the described treatment enabled an increase of at least 3 days (20%) of the products shelf life, reducing food waste and contributing to food security.

## 1. Introduction

The deterioration of food products can occur at any stage of the supply chain, especially when they reach their “best before” date. Different approaches have been developed to find solutions to food loss, allowing food products to last longer. Among the studies on preservation techniques, solutions involving different packaging processes (vacuum, modified atmosphere, etc.), cooling specifications, freezing, sterilization, among others, confirm the role of such technologies to provide new solutions. There are also opportunities in other aspects of global food wastage, such as extending the shelf life of fresh food and the prevention of its deterioration even by just one day. It is imperative to identify these solutions and make these changes simple and safe throughout the supply chain [1,2].

Consumption of bell peppers (*Capsicum annuun*) has increased, either as major ingredients in salads or as side dishes. They give color, flavor and pungency to recipes and, simultaneously, they have desirable sensory properties [3,4,5]. Peppers are rich in bioactive compounds (phenols, flavonoids, anthocyanins, carotenoids and vitamins A and C), which are correlated to health benefits and protection against diseases like cancer, cataracts and macular degeneration [6,7,8]. However, the maturation stage and genotype, the processing operations and storage conditions of the products can affect their phytochemical content [9]. Peppers are sensitive to chilling temperatures, not being suitable for long, cold storage periods. Adverse storage conditions (T = 5 ± 2 °C) promote water loss and surface pitting, shrinkage, softening, physiological disorders and/or fungal infections [6].

A great research effort has been made regarding the preservation of vegetables by modified atmospheres. Most of these studies have involved whole samples packed in passive MAP or long storage periods in controlled atmospheres [10,11,12,13]. Studies involving fresh-cut products, preserved in active MAP during large periods and supported by a broad set of parameters are scarce. More research on gas combinations and their effects on senescence and biological activity of the products are needed [14,15]. Moreover, maintaining the optimum range of temperature and moisture during postharvest handling, the use of high CO_2_ atmospheres might be a determinant factor to maintain fresh produce quality. For this reason, this work aimed to evaluate the changes in several physicochemical parameters and bioactive compounds content, as well as the evolution of texture, microbial growth and sensory properties of green (Gp), red (Rp) and yellow (Yp) peppers, minimally processed and packed in MAP (10% O_2_ and 45% CO_2_), seeking the extension of the “best before” currently adopted by industry.

## 2. Results

To evaluate changes in peppers’ quality stored as described in Section 4, several parameters concerning the physicochemical, textural, microbiological and sensory characteristics of the products were measured in order to obtain a picture, as complete as possible, of the product’s evolution.

### 2.1. Packaging and Gas Composition

The evolution of CO_2_ and O_2_ levels inside packages during storage of Gp, Rp and Yp in MAP at 5 °C was monitored from processing day (day 0, control) to the 17th day of storage (Figure 1).

The initial O_2_ concentration was 10%, but after five days of storage all packages had around 15% O_2_ (Figure 1a). Its concentration in all products and sampling times did not drop to low levels avoiding harmful physiological reactions.

The CO_2_ percentage along the evaluation period decreased gradually (Figure 1b) and kept near the recommended level at the end of the storage period, between 2 and 5% [6,16,17]. At the end of the study CO_2_ levels attained a minimum of 5.3% in Gp, 5% in Rp and 2.0% in Yp samples. This reduction in CO_2_ levels is due to diffusion across the film and to dissolution in the product. Similar final CO_2_ levels were obtained by Manolopoulou et al. that found a 14-day shelf life for fresh-cut green bell peppers [6].

Transpiration did not occur at a concerning level as no condensation inside the package was observed. Only in the latest sampling periods some exudates were observed.

### 2.2. Moisture and Ash

Water loss generally results in a reduction of fresh weight causing degradation of appearance and loss freshness and firmness [18]. The results (Figure 2A) showed no significant changes in moisture content over the storage time (*p* > 0.05) with losses of 1–2%, except on the 5th day (Gp and Rp) and the 10th day of storage (Yp), reaching means of 3–4% (*p* < 0.05). Concerning ash content (data not shown), it was quite stable over storage time and no significant changes were observed (*p* > 0.05).

### 2.3. Titratable Acidity and pH

The organic acids are among the compounds considered in this parameter, which greatly influence taste, aroma, color and overall stability of the vegetables. Figure 2B,C show the evolution of the values of titratable acidity and pH along the storage time.

Titratable acidity in Gp samples did not vary significantly over storage time in comparison to the control (processing day; *p* > 0.05). Concerning Rp samples, titratable acidity varied over storage time showing an increase on the 5th day of storage (*p* < 0.05; post-hoc Tukey HSD), followed by a decrease and stabilization. In the case of the Yp, titratable acidity decreased significantly (*p* < 0.05) on the 5th day of storage but stabilized during the remaining storage time. Comparing titratable acidity values among the three types of peppers, Yp presented the highest mean values (0.15 g citric acid/100 g fresh weight) followed by the Rp (0.1 g citric acid/100 g fresh weight) and Gp (0.06 g citric acid/100 g fresh weight). As it can be observed in Figure 2C, Rp presented the lowest pH values (5.1, mean value), quite stable over time. The pH of Yp (5.3, mean value) was slightly above and also stable over the study time. Gp presented a small pH variation after the 10th day of storage (6.0, mean value).

### 2.4. Texture

Significant changes (*p* < 0.05) in fresh cut bell peppers firmness were observed only in the first 5 days of sample storage (Figure 3A). After that, the rupture force decreased to initial values (9–10 N) and tended to be constant along the remaining days.

### 2.5. Color Evaluation

The color parameters of green, red and yellow fresh-cut bell peppers in MAP treatment from processing (day 0, control) to the 17th day of storage are presented in Figure 3B–D and tabulated in Table 1.

The lightness value (*L**) of the samples ranged from 33.70 to 34.58 (Gp), 38.79 to 37.61 (Rp) and 61.39 to 57.24 (Yp), showing that *L** values in MAP treatment were not significantly reduced in all samples by the end of storage period (day 17). The *C** values (intensity of color) were reduced in 12, 8 and 17% in Gp, Rp and Yp samples, respectively. The saturation *C** ranged from 16.37 to 50.22 according to the different types of bell peppers. Chroma is calculated by squaring *a** and *b** values, making the negative *a** (green color) into a positive value, masking its influence. Because *Chroma* reflects color purity or saturation, it could be a good indicator of consumer acceptance to red bell peppers, taking into account its non-influence on positive *a** values. Likewise, vegetables with different colors might have a similar distinctive *hue angle* as found in green, red and yellow peppers (Table 1). The initial hue angle (*h**) for Gp, Rp and Yp was 116.87, 41.56 and 84.31, respectively, and presented non-significant variations in MAP samples throughout the storage period. All MAP samples had similar *h** values at the end of the storage, indicating good retention of the initial color, which may be due to limited dehydration and/or limited pigments degradation. In all cases, no-significant or absence of *L** value reduction (<7%) and *h** retention throughout the MAP storage indicated the retention of the initial color, showing that these were the best color quality parameters especially for green and yellow fresh-cut bell peppers.

### 2.6. Total Phenol Content

According to Figure 4A, Gp samples have higher TPC (initial value: 280 mg GAc eq/L ext or 12.6 g GAc eq/kg fresh Gp) than Rp and Yp (initial values: 240 mg GAc eq/L ext or 10.8 g GAc eq/kg fresh Rp) and 260 mg GAc eq/L ext or 11.7 g GAc eq/kg fresh Yp, respectively), although with no significant difference (*p* > 0.05) along the storage period. These results are similar to the ones reported by Zhang and Hamauzu [9].

### 2.7. Total Flavonoid Content

The evolution of TFL content is presented in Figure 4B. Samples maintained their levels (initial values: Gp, 200 mg/L ext. or 9 g/kg fesh Gp; Rp, 80 mg/L ext or 3.6 g/kg fresh Rp; 50 mg/L ext or 2.25 g/kg fresh Yp) without changes along the storage period (*p* > 0.05). Yp are the poorest samples in these bioactive compounds.

### 2.8. Anthocyanin Content

Although some small changes were observed during the storage of peppers (Figure 4C), these changes seemed to be random, with no apparent trend. In the case of de Gp (initial value: 108 mg/L ext or 4.86 g/kg fresh Gp) and Rp (initial value: 110 mg/L ext or 4.95 g/kg fresh Rp) small differences are graphically observed (Tukey HSD *p* < 0.05 for samples with 10 and 14 days of storage, respectively), and no significant alteration for Yp (initial value: 100 mg/L ext or 4.5 g/kg fresh Yp) was registered (*p* > 0.05).

### 2.9. Carotenoid Content 

The carotenoid content in Yp and Gp is lower than in Rp not following the tendency described by Zhang and Hamauzu [9] but in accordance to Navarro et al. [19], who stated that Rp presented higher values of biologically-active carotenoids. The carotenoid evolution along the study (Figure 4D) is also different in each case. Rp samples presented small changes over the storage time probably due to the advanced maturation stage. Changes in Yp and Gp samples were not observed (*p* > 0.05), except in samples with five days of storage, that had slightly high values, as happened with other parameters.

### 2.10. Antioxidant Activity and Bioactive Compounds Evaluation

Bioactive compounds (total phenol, flavonoid, anthocyanin and carotenoid) and antioxidant activity by DPPH (EC_50_) were evaluated and any trend in the evolution of these parameters could be pointed out. The EC_50_ for the three pepper samples (initial values: Gp, 1.1 mg/ml; Rp, 1.16 mg/ml; Yp, 1.7 mg/mL) and over storage time were not significantly different (*p* > 0.05), exception made to the latest sampling days for Gp and Rp (both with 1.7 mg/mL, 17th day) visible on the plot (Figure 4E). This is an important result as losses in total antioxidant activity are negligible.

### 2.11. Microbial Growth Control

The results obtained from microbial growth control are presented in Figure 5. Total microorganisms count at 30 °C remain below the admitted limits, according to microbiological criteria of the HPA Guidelines for Assessing the Microbiological Safety of Ready-to-Eat Foods Placed on the Market, established by the Health Protection Agency [20]. It is worth noting that total microorganisms’ count at 30 °C was only used to help deciding if samples were safe for sensory studies. The authors are aware that these microbiological parameters alone are not enough to assure microbiological safety.

### 2.12. Interplay of Parameters

Due to the high number of parameters and individual variations observed, a multivariate analysis was performed to look for main data structures and possible trends (Figure 6). PCA with predictive biplots was chosen since it enables to carry out interpretations based on initial parameters and respective correlations. To interpret the biplot displays, a straight and orthogonal line is drawn from a sample point to a variable axis, and the value of the variable in that sample is read directly in the display. In this way, principal components do not need to be interpreted and judgments are based on the original data.

Figure 6 shows that over time the evaluated parameters presented only slight variations. The results corresponding to samples with five days of storage (Gp1, Rp1 and Yp1) deserve some mention, since they generally presented different values from samples on processing day (day 0, control).

The analysis showed the existence of three important data structures visible in Figure 6a: titratable acidity, ash, *b** and *L** which are highly correlated (respective axes are almost collinear); pH and moisture content highly correlated between themselves; a lower correlation between these two structures can also be inferred; a third structure represented the correlation between CFU, *a** and TC. In Figure 6b it becomes clear that TXT is not correlated with other parameters, behaving as a data structure by itself.

In the display of Figure 6b, it is observed that on variable TXT (texture/firmness) sample Rp1 projects to 23 N while the other Rp samples project to between 7 and 17 N; sample Gp1 to 17 N and the other Gp samples to between 7 and 11 N; and finally Yp1 to 16 N, while the others project to between 9 and 12 N. As it is seen this kind of interpretation is very easy and helpful. These results are in agreement with the discussions presented on physicochemical parameters and according to Figure 2, Figure 3 and Figure 4.

Concerning samples other than samples with five days of storage, no trend was identified. These samples are projected in the graph in the neighborhood of the fresh samples, with a negligible distance. Most of the analyzed parameters showed no significant differences between the storage times under study (*p* > 0.05).

Generally, moisture loss is followed by an increase in pH and a decrease in acidity. Strong correlations were observed between the ash content and titratable acidity, presumably due to combinations between minerals such as calcium, phosphorus, and potassium with the organic acids which could influence the buffering capacity [21].

Color parameter *b** is negatively correlated with moisture content and pH, although, as it is seen in Figure 6a, this correlation is relatively small as represented by an increased angle between the respective axes. As expected, PCA differentiates Gp from Rp in terms of the parameter *a**, placing them in opposite ends of the *a** axis drawn on the PC1 vs PC2 plot (Figure 6a). Similarly, Yp samples project onto the *b** axis edge corresponding to the highest values of *b**. It is also seen that the units projecting to higher values of *a** also present lower pH values. These lower pH values may be indicative of some inactivation of cell wall enzymes, which may lead to some loss of firmness [22,23]. Concerning texture, in the plot of PC1 vs PC3 (Figure 6b) it becomes clear that at the first sampling time samples presented higher values of firmness.

PCA reveals no significant changes in TPC content over time, except in the case of Rp and Gp samples on the 5th day, for the same reasons explained before. In the case of flavonoids, PCA only gives them relevance on the plane PC2 vs PC3. The corresponding axis (TFL) does not appear on the plane PC1 vs PC2 because it does not follow the trends of other variables in this plot and does not contribute to separate varieties or sampling times. These results show that the preservation technique allows conservation and it is able to maintain flavonoid content, an important conclusion from the nutritional point of view.

### 2.13. Sensory Evaluation

PCA with VARIMAX rotation of sensory data (Figure 6c) shows some strong correlations between attributes, such as flavor typicity (A13) and taste intensity (A14). This analysis also removes from the plot the axis of the attributes with a mean standard predictive error higher than 0.5. These attributes are A3, A5, A6, A9, A10, A12, A15, A16 and A17 related to all the categories of characteristics evaluated (appearance, texture, mouth feel, aroma and taste). This illustrates the difficulty experienced by the panel during the evaluation process, due to the difficulty in pointing out differences between samples or finding out problems in the samples, when comparing to standard (fresh product) always present in every session. The majority of the samples are projected together close to the center of the sensory evaluation scale in the neighbour of point 7 corresponding to the classification attributed to the fresh samples.

The PCA plot PC1vs PC2 (Figure 6c) shows that Rp samples with 10 (Rp3) and 17 (Rp4) days of storage stand out from the remaining. The former has high values for attributes A15 and A2, and low values for A6 and A7. The latter is more extreme than Rp3 and shows very high values in A4 and low values in A8. An increasing sour taste (A15) in Rp3 samples and an excessive brightness (A4) and surface moisture (A1) revealing an abnormal release of exudative liquids in Rp4 are the characteristics that allowed to consider these samples out of the remaining. However, sour taste is considered typical in bell peppers and as it can be seen in the plots, the A15 axis has the same orientation and is close to A13 and A14. Gp samples with 17 days of storage (Gp4) are also judged as slightly different from the others, mainly due to low values in A8, although not as extreme as Rp4. The separation of these samples from the main group occurred because the panelists found them softer and also with usual brightness and surface humidity. Apart from these differences, in judges’ opinion, samples were quite similar to the standard and no major problems were found.

## 3. Discussion

Combining the right temperature and an adequate packaging film is now widely recognized as an efficient preservation method to overcome senescence. The literature review data of minimally processed fresh bell peppers shelf lives stored at different packaging conditions are reported in Table 1. The shelf life studies for bell peppers described in the scientific literature range from 10 to 49 days, depending on the initial characteristics of the samples, treatment, specific packaging and storage conditions. Among these studies, the shelf life of whole bell peppers, with or without MAP, ranged from 12 to 28 days (Table 2). The best results were obtained by Renu and Chidanand, who used a diffusion channel system and pressurized argon treatment (28 days, 8 °C, 95% RH) for whole green bell peppers [24]. Sahoo et al. also used a MAP for whole green bell peppers (9% O_2_; 4% CO_2_, 4–6 °C, 45% RH) obtaining a significant shelf life of 20 days using a polypropylene (PP) film and vacuum packaging [25].

All studies using MAP and fresh-cut samples used green bell peppers, with the exception of Sharma et al., that used red bell peppers, finding a shelf life of 14 days (21% O_2_, 0.3% CO_2_, 7 °C, 94% RH) in a specific type of packaging material [26]. Among these studies, the results obtained by Singh, Giri and Kotwaliwale (2014) should be highlighted [10]. The authors found an unusual shelf life of 49 days (4.5% O_2_, 7.5% CO_2_, 8 °C, 95% RH) using a specific packaging system specially designed by them with permeable PP polymeric films.

Experimental designs that are similar to the one proposed in the present study showed shelf lives ranging from 10 to 14 days for fresh-cut green bell peppers, according to atmospheric conditions and types of packaging (Table 1). In those studies, initial O_2_ levels ranged from 2 to 22% and CO_2_ levels from 8 to 15%. This initial CO_2_ level range is lower than the proposed in the present study which findings will be discussed below.

The mean values for titratable acidity and pH for both Rp and Yp showed small differences among them. Accordingly Castro et al. also reported that Rp presented higher titratable acidity values than Gp, probably due to the different organic acids in their composition [22]. The pH values determined in this study are in accordance with other authors [31,32]. Despite a slightly higher mean absolute pH value for Yp, an indirectly proportional relationship between these two parameters was observed. This can be related to a similar composition of organic acids for Rp and Yp. In the case of Gp, due to a probable distinct organic acid composition, a clear indirect proportional relationship was observed but with a reversal of absolute values. The most significant variations were observed for Rp (titratable acidity, 5th day) and Gp (pH, 10th day), showing a differentiated period of adaptation to the new environmental conditions, whose values stabilize in function of time.

Significant changes in fresh cut bell peppers firmness were observed only in the first 5 days of sample storage which might be due to the washing of the fresh-cut pepper slices, which improve firmness probably by removing, from the cut surfaces, some solutes and stress-related signaling compounds [12,33]. Moreover, these results can be due to the initial shock with high CO_2_ concentrations, leading to the firmness changes in the initial days of storage.

Modified atmosphere (reduced O_2_ and elevated CO_2_) can extend the post-harvest shelf life of fresh-cut plant products by reducing respiration rates and ethylene production. These atmospheres along with low temperature storage minimize metabolic activity delaying enzymatic browning and retaining sensory characteristics [34]. The *L**, *a**, *b**, *C** and *h** values were used to monitor the changes in pepper color during storage. Small changes were observed, following different patterns depending on the type of bell pepper. Darkening, reflected in a decrease in absolute *L** values, occurred at a higher extent in Yp when compared to Gp and Rp, probably reflecting enzymatic activity of polyphenol oxidases due to mechanical shock and the easy browning in this type of bell pepper [12,24]. These enzymes, which occur in many plants together with phenolics, catalyze the oxidation of mono-, di-, and polyhydric phenols to quinones, regulating the enzymatic browning process [35]. Results from non-parametric analysis of color (Kruskal-Walis chi-squared test) revealed no significant differences (*p* > 0.05) over time (Figure 3B–D). The correlation profile of the studied variables showed a high correlation of *L** with titratable acidity, which in turn was highly correlated with pH and this one influences the enzymatic activity as referred above. In all cases, for *a** and *b** values, no significant differences were observed during the experiment time (*p* > 0.05).

Specifically, green and yellow bell pepper samples showed *a** mostly negative values, possibly due to the high content of pigments such as chlorophylls and carotenoids. Analyzing these results, it is possible to infer that different colors have simultaneously influenced the behavior of green and yellow samples. This can be justified since chlorophyll is degraded from green to colorless compounds at the same time that carotenoids are synthesized from phytoene, a colorless precursor to ξ-carotene, lycopene, β-carotene, xanthophylls and hydroxylated carotenoids in a parallel biosynthetic pathway [36].

Retention of the color in fruits and vegetables due to limited pigment degradation is probably consequence of the establishment of steady-state microenvironment inside the package, with elevated CO_2_ and/or depleted O_2_ concentrations. The color activity concept and the *L**, *a** and *b** color system are useful tools for helping understand the contribution of individual pigments to the overall color of vegetables and address an important quality parameter, especially fruit decay and marketability [10].

Bioactive compounds play a protective role against harmful free radicals and peppers are known to be important sources of these antioxidant compounds [10]. Bell peppers are rich in ascorbic acid, a vitamin with many biological activities in the human body, being an important nutritional quality factor in some horticultural crops. The vitamin C content in fruits and vegetables is influenced by various factors such as maturity, genotype, preharvest conditions, harvesting methods and postharvest handling procedures. The loss of vitamin C after harvest, for example, can be reduced by storing fruits and vegetables in reduced O_2_ and/or up to 10% CO_2_ atmospheres; higher CO_2_ levels can accelerate vitamin C loss [37].

Generally, modified atmospheres during storage decrease physiological and chemical changes of fruits and vegetables. The combined effect of low temperatures (0 and 5 °C) and MAP can be beneficial to retain the initial ascorbic acid, as reported by Manolopoulou et al. (2010) [6]. Other studies using MAP and temperatures from 0 to 8 °C showed that the ascorbic acid content did not change significantly [10,14,28].

Piga et al. in a work with minimally processed cactus pear fruits, expected a high decrease of ascorbic acid content [38]. Nevertheless, such situation was not verified, inferring the authors a probable protection promoted by the ascorbate-sparing effect of polyphenols. The authors also reported no significant decrease in Trolox equivalent antioxidant capacity (TEAC) of the samples during storage at 4 °C. The correlation between TEAC values and ascorbic acid content in fruits presented a high value, confirming that ascorbic acid was the main antioxidant. The contribution of the polyphenols was also determined by the authors, reporting that the correlation factor was also consistent but lower than that of ascorbic acid. In the present study, the ascorbic acid content analysis was not performed but the antioxidant activity was used as a reference for a correlation between them.

## 4. Materials and Methods

### 4.1. Samples

Freshly harvested green (Gp), red (Rp) and yellow (Yp) bell peppers (*Capsicum annuun* L. var *annuum*) obtained in local farmers from Póvoa do Varzim (Portugal) were the study samples. Old stems and dark spots were removed before cutting. Pre-washing of the samples was carried out with potable water at room temperature and disinfection with a 0.3% Verclor (5 min, room temperature, Soro Internacional, SA, Zaragoza, Spain), followed by rising washing. Peppers were dried after the sanitation procedure and cut into transversal slices of 2 cm with a sharp blade knife to avoid extra mechanical damage.

### 4.2. Minimally Processed Vegetables

About 200 g of each sample were packed in Krehalon MLF40 (PA/PE) bags (Kureha Corporation, Tokyo, Japan) and sealed using a VacuumMit P20 (Caso, Arnsberg, Germany) gas flushing and thermosealer coupled to an A/SMAP Mix 8000EL gas mixer (PBI Dansensor, Ringsted, Denmark). Samples were prepared in triplicate for each sampling day. 

Packaging atmosphere was optimized in a previous study and obtained by an active way with initial gas composition of 10% O_2_/45% CO_2_ (O_2_ and CO_2_ transmission rates of 90–130 cm^3^ m^−2^ × day and 750–850 cm^3^ m^−2^ × day, respectively), with N_2_ balanced to 100%; 20 ± 2 °C; 60 ± 1% RH. All minimally processed samples were stored at 5 °C [39]. Sampling for overall quality evaluation was made at the beginning of the experiment (processing day, Control) and on the 5th, 10th, 14th and 17th days of storage. On each sampling day, packaging gas composition was monitored with a gas analyser (O_2_/CO_2_ CHECKMATE II, PBI Dansensor). For physicochemical analysis (except color and texture), samples were grinded in a knife mill; for bioactive compounds analysis they were lyophilised and stored at −20 °C until further analysis.

### 4.3. Moisture and Ash Contents

Moisture and ash contents were determined according to AOAC methods [40]. About 3 g of grinded and homogenized samples were dried at 103 ± 2 °C (UTG, Heareus Instruments, Schwerte, Germany) to constant weight. A similar amount was incinerated in a muffle at 550 °C until complete carbonization, cooled and weighed. Determinations were performed in triplicate and the results expressed as percentage of total weight.

### 4.4. Titratable Acidity

For titratable acidity evaluation the grinded peppers (25 g) were homogenized in 250 mL of water and titrated with 0.1 N NaOH. The results were expressed as g citric acid/100 g fresh weight [40].

### 4.5. pH Analysis

Evolution of pH values was monitored using a FC232 electrode coupled to a potentiometer HI99163, HANNA Instruments (Woonsocket, RI, USA) according to AOAC methods [40].

### 4.6. L*, a*, b*, C* and h* Color

Color measurements were carried out with a MINOLTA CR300 (Konica Minolta, Ramsey, NJ, USA), using the Hunter *L**, *a**, *b** system. Measurements were performed longitudinally joining four pepper slices in a way that the diameter measuring head (8 mm) was completely surrounded by the product surface. CIE *L**, *a**, *b** readings were calibrated against a standard white plat and measurements repeated 10 times for each sample. The degree of redness and greenness was measured by *a**, as +*a** and −*a**, respectively. The degree of yellowness and blueness was measured by *b**, as +*b** and −*b**, respectively. The measured values (*a** and *b**) were used to estimate chroma values, *C** = (*a**^2^ + *b**^2^)^0.5^ and hue angle degrees *h** = arctan (*b**/*a**). Hue angle values greater than 90° correspond to intense green color while values close to 90° indicate yellow color. Chroma defines the color intensity or purity of the hue. Values close to 0 correspond to neutral colors and values close to 60 to bright colors [10]. Depending on the sampling scheme and accuracy, the *L**, *a**, *b**, *h** or *C** values can provide a satisfactory color description [6].

### 4.7. Texture Analysis

Firmness, a texture parameter defined as the maximum force exerted in compression, was measured as the puncture force in outer cortex with a Texture Analyzer TA-XT2i (Stable Micro Systems Ltd., Goldalming, UK; n = 20) equipped with a 5 N load cell and a 2 mm diameter cylindrical stainless-steel probe [41]. Compression was performed with speed of 1 mm·s^−1^ and penetration depth of 0.7 mm.

### 4.8. Extraction of Bioactive Compounds

Extracts were prepared according to Porter [42] with some modifications. About 1.0 g of lyophilized samples was extracted by agitation (5 min) with 15 mL of 80% methanol using a vortex shaker (WZ-04726-01, Cole-Parmer, St. Neots, UK) and filtered through Whatman n^o^ 4 filter paper (GEHealthcare, Chicago, IL, USA) [43]. The residue was extracted 2-fold again with 15 mL of 80% methanol and filtrates were centrifuged at 3500 rpm (25 °C) for 45 min. Supernatants were collected and stored at −20 °C till analysis.

### 4.9. Total Phenolic Content (TPC)

Total phenolic content was determined by the Folin-Ciocalteu method [44]. The results were expressed as milligrams of gallic acid equivalent (GAc eq) per liter of extract. Sample extracts (500 μL) were added to 2.5 mL of Folin-Ciocalteu reagent and 2.5 mL of a 7.5% NaCO_3_ solution (dilution factor 1:10). Absorbance was measured at 765 nm with a Synergy HT Microplate Reader (BioTek Instruments, Inc., Winooski, VT, USA) after 15 min incubation time at 45 °C followed by 30 min at room temperature. An analytical curve of gallic acid with concentrations ranging from 5 to 100 ppm was used as standard and the analyses performed in triplicate.

### 4.10. Total Anthocyanin (ANT)

ANT was determined following Li et al. [45]. Sample extracts (2.5 mL) were dissolved in 25 mL of methanol, placed in an ultra-sound bath for 10 min and heated at 40 °C during 30 min. Absorbance (Abs) was read at 528 nm and anthocyanin content calculated using the following equation:(1)$$\mathrm{mg}\;\mathrm{Cyd-3-Glu}\;\mathrm{eq}/L\;\mathrm{ext}=\frac{Abs\times MW\times DF\times 10^{3}}{\varepsilon \times l}$$
where Abs = Absorbance; MW (molecular weight) = 449.2 g/mol for cyanidin-3-glucoside (Cyd-3-Glu); DF = dilution factor (1:10); l = path length in cm; ε is the molar extinction coefficient (26900), in L mol^−1^ cm^−1^, for cyd-3-glu; and 10^3^ = factor for conversion from g to mg. Results were expressed in mg Cyd-3-Glu eq/L extract solution.

### 4.11. Total Flavonoid Content (TFL)

TFL content of extracts was determined using a methodology previously described by Costa et al. [46]. Catechin standard solutions in the range of 0–400 ppm were prepared for the analytical curve. A white essay was used to calibrate the zero in the equipment and absorbance was measured at 510 nm.

### 4.12. Total Carotenoid (TC)

Total carotenoid quantification was performed as described by Yuan et al. [47]. About 1.5 g of lyophilized samples was mixed with 40 mL of pure acetone, stirred for about 15 min, protected from light, and filtered. Petroleum ether (30 mL) was used to wash the residue and repeated twice. The filtrate was transferred to an ampoule and washed for several times with distilled water. The upper phase was collected, and the volume measured. TC content was determined by spectrophotometry (Spectrophotometer UV-1800, Shimadzu, Kyoto, Japan), at 450 nm ($A_{1\mathrm{cm}}^{1\%}$ = 2592), using petroleum ether as the blank. The results were expressed in μg of carotenoids/g fresh sample.

### 4.13. Antioxidant Activity-DPPH^•^ Assay

The evolution of the antioxidant activity over storage time was determined by DPPH^•^ radical scavenging according to Brand-Williams et al. with modifications [9,42,43,48]. A DPPH^•^ solution (0.1 mM in 80% methanol) was added to the extracts and a control prepared with distilled water instead of the extract. Absorbance was measured at 521 nm with a Synergy HT Microplate Reader (BioTek Instruments Inc., Winooski, VT, USA). Samples and controls (25 μL) were applied into each well followed by 200 μL of DPPH solution. Radical scavenging activity (RSA %) was calculated as the percentage of discoloration of DPPH solution, using the following equation:(2)$$\mathrm{RSA}\;\%=\frac{Abs\;\left(control\right)-Abs\left(sample\right)}{Abs\;\left(control\right)}\times 100$$

The antiradical activity is defined as the amount of sample needed to reduce the initial DPPH^•^ concentration to 50% (EC_50_). This concentration was obtained from the plot of RSA % as a function of sample concentration.

### 4.14. Total Microorganism Counting

Microbial evaluation was performed as indication of the deterioration extent based on total bacterial aerobic colonies count technique at 30 °C. Sampling was done in aseptic conditions according to ISO 6887 [49]. Small portions from different zones of the package were used to prepare the test sample for microbiological analysis. A mother suspension (10^−1^) was prepared with 10 g and introduced in a sterile “Stomacher” bag, mixed with 90 mL sterilized Triptone salt solution, and homogenized for 1 min in a Stomacher 400 (BA7021, Seward, West Sussex, UK). Decimal dilutions were prepared (10^−3^) and 1 mL of each dilution was pour-plated on Plate Count Agar (PCA) medium (Oxoid, Basingstoke, Hampshire, UK) and incubated at 30 ± 1 °C for 72 ± 3 h, as recommended by ISO 4833 [50]. Results were expressed in log Colony Forming Units (CFU) g^−1^ of sample.

### 4.15. Sensory Evaluation

The sensory evaluation followed the QDA^®^ methodology (Tragon Corporation, San Francisco, CA, USA), as described by several authors and implemented according to Alves and Oliveira [51,52,53]. The methodology involves a simultaneous selection of judges and attributes. Before the evaluation sessions, panel judges focused on the typical characteristics of fresh bell peppers and on the presence of defects caused by senescence. Fresh samples (prepared in the moment) and samples kept at 5 °C for different periods of time were used by judges in several discussion/training sessions, in order to understand the course of deterioration and its effects on sensory quality. A list of attributes was decided by consensus, containing the following attributes: general appearance (including surface moisture (A1), typical color (A2), browning (A3), brightness (A4), texture/shape (A5); texture measured with forks (including elasticity (A6), cohesiveness (A7), hardness(A8)), aroma typicity (A9), and intensity (A10)); texture during mastication (firmness at first bite (A11), firmness at second bite (12)), flavor and taste (including flavor typicity (A13), taste intensity A(14), sourness (A15), bitterness (A16), sweetness (A17)). Thirteen-point continuous scales were used for all attributes.

Products were evaluated by trained judges (*n* = 15, age ranging from 20 to 55 years). All samples were presented to the judges in small white plates with around 5 g each (2–3 thawed slices). In all sessions, fresh cut vegetables were presented as standards. Judges were instructed to start any session by tasting the standard and consider point 7 of each attribute’s scale as the standard’s magnitude perceived for that attribute. In this way, in all sessions and for all judges, evaluations were made against a fresh standard, reducing randomness in judgments.

### 4.16. Statistical Analysis

Evaluation of the main characteristics and the way they changed with time was carried out with Principal Component Analysis (PCA) and predictive biplots, using the AutoBiplot.PCA function written in R (The R project for Statistical Computing; http://www.r-project.org/) [54]. Biplot axes, representing initial variables equipped with appropriate measurement scales, were automatically drawn in the biplot displays using a mean standard predictive error (MSPE) of 0.5. R was also used to evaluate the significance of observed differences among several groups through parametric univariate ANOVA tests and Tukey HSD post-hoc tests, as well as with their corresponding non-parametric counterparts, Kruskal-Wallis and Wilcoxon/Mann-Whitney tests [55]. All graphs were produced with R programs specially built by the authors.

## 5. Conclusions

Green, red and yellow fresh-cut bell peppers of *Capsicum annuun* L. var *annuum* were stored at 5 °C for 17 days employing packaging film bags (Krehalon MLF40-PA/PE) in a modified atmosphere (10% O_2_ and 45% CO_2_). Bell peppers exhibited trivial overall quality degradation. The tested storage conditions and packaging film kept the in-package O_2_ concentrations above 10%. The percentage of CO_2_ used was higher than the usually described for controlled atmosphere for MAP (5–15%). Although the initial CO_2_ concentration used was high (45%), causing an initial shock, the final levels of CO_2_ were between the recommended levels (2–5%) due to the porosity of the film used, allowing higher gas exchange rates and the release of CO_2_ excess, thus minimizing possible deleterious effects. The effect of the initial high CO_2_ percentage powers the barrier effect of MAP by slowing down metabolic reactions in green, red and yellow fresh-cut bell peppers. Small physical and chemical changes were observed during the storage period evaluated (17 days) and changes tended to stabilize, with no significant differences between the processing day (control) and sampling periods till the end of the study. Sensory analysis confirmed the results with experienced panelists not being able to discriminate products between sampling periods.

In light of the hurdle technology, severe changes in the atmosphere gas composition, low storage temperatures and the use of a package composed of a polymer with higher permeability than those commonly used in the industry allowed the deceleration of quality loss. From the previous findings it was concluded that cold storage of green, red and yellow fresh-cut bell peppers (var *annum*), packaged in film bags (Krehalon MLF40 - PA/PE) at 5 °C, can be stored up to 17 days with minor quality degradation, increasing their current commercial shelf life in at least 3 days (20%) and reducing food waste. To the best of our knowledge this is the first report evaluating a MAP treatment simultaneously in three different fresh-cut bell peppers (green, red and yellow) and using an initial shock with high CO_2_ concentrations.

## Figures and Tables

**Figure 1 molecules-25-02323-f001:**
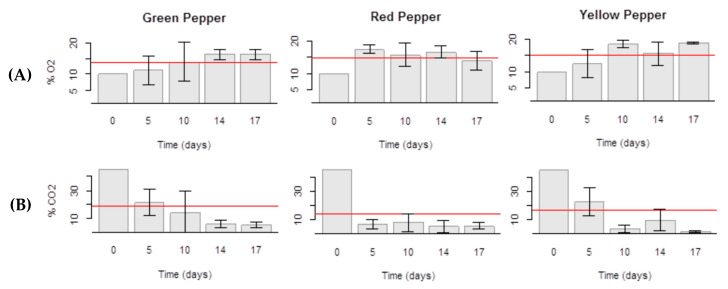
Evolution of the atmosphere gas composition inside the package ((**A**) % O_2_; (**B**) % CO_2_) of fresh-cut bell peppers (green pepper; red pepper; yellow pepper) stored under an initial MAP condition of 10% O_2_ and 45% CO_2_ throughout 17 days at 5 °C. The statistical mean is represented by the horizontal line.

**Figure 2 molecules-25-02323-f002:**
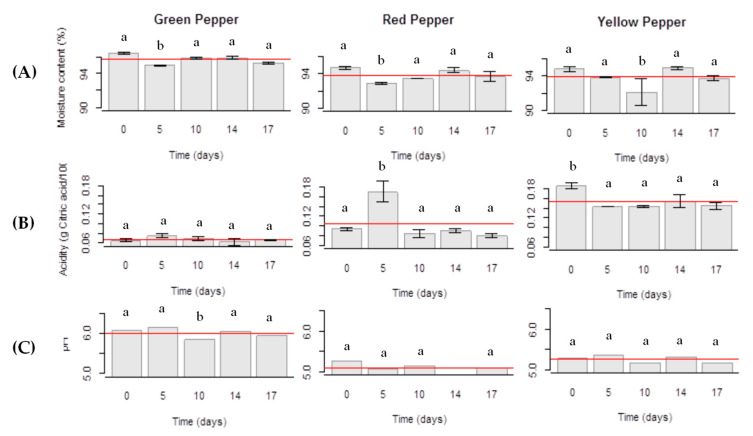
(**A**) Moisture content, (**B**) titratable acidity and (**C**) pH values of fresh-cut bell peppers (green pepper; red pepper; yellow pepper) stored under an initial MAP condition of 10% O_2_ and 45% CO_2_ throughout 17 days at 5 °C. a—no significant differences (*p* > 0.05); b—significant differences (*p* < 0.05). The statistical mean is represented by the horizontal line.

**Figure 3 molecules-25-02323-f003:**
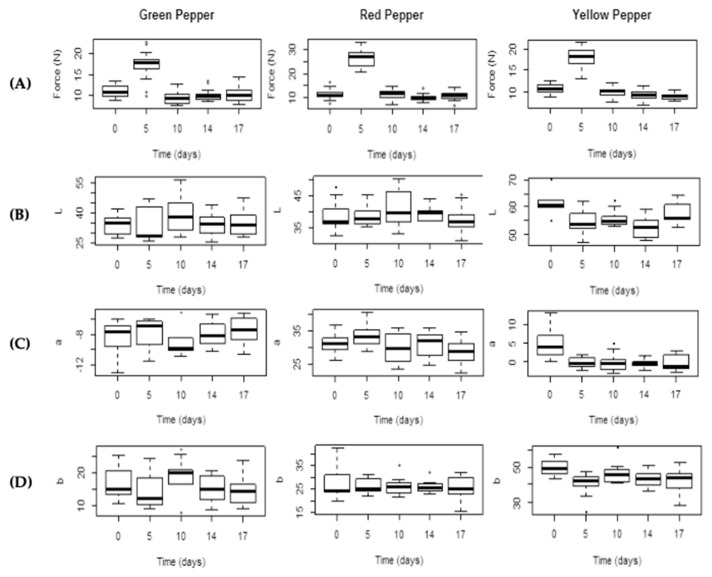
(**A**) Instrumental firmness; (**B**–**D**) *L**, *a** and *b** color values of fresh-cut bell peppers (green pepper; red pepper; yellow pepper) stored under an initial MAP condition of 10% O_2_ and 45% CO_2_ throughout 17 days at 5 °C.

**Figure 4 molecules-25-02323-f004:**
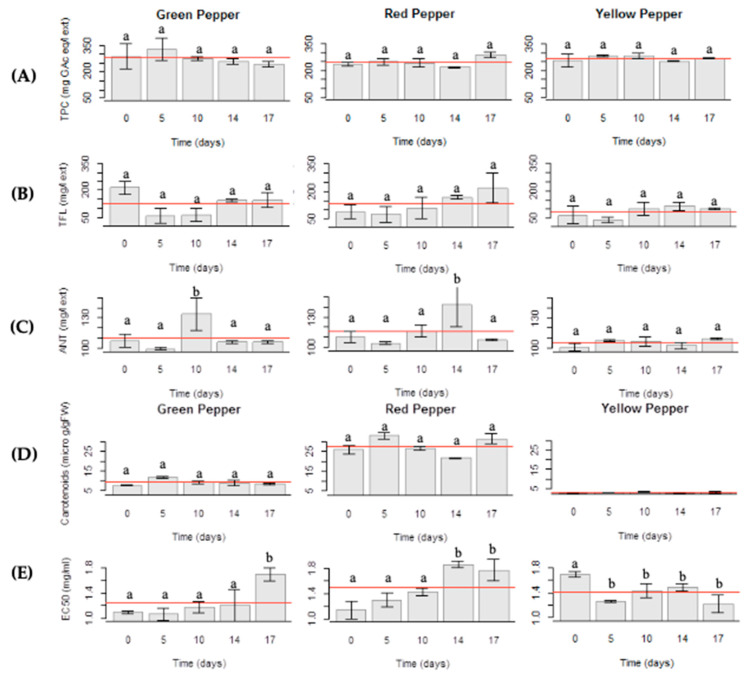
(**A**) Total phenol content (TPC); (**B**) Total flavonoid content (TFL); (**C**) Total anthocyanin content (ANT); (**D**) Total carotenoid content and (**E**) Antioxidant activity (EC_50_) in fresh-cut bell peppers (green pepper; red pepper; yellow pepper) stored under an initial MAP condition of 10% O_2_ and 45% CO_2_ throughout 17 days at 5 °C. a—no significant differences (*p* > 0.05); b—significant differences (*p* < 0.05). The horizontal line represents the statistical mean.

**Figure 5 molecules-25-02323-f005:**
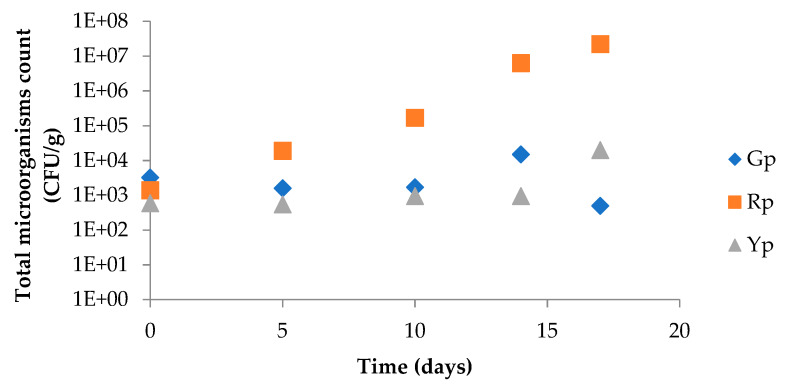
Total microorganisms count (log CFU g^−1^) at 30 °C of fresh-cut bell peppers (green pepper; red pepper; yellow pepper) stored under an initial MAP condition of 10% O_2_ and 45% CO_2_ throughout 17 days at 5 °C.

**Figure 6 molecules-25-02323-f006:**
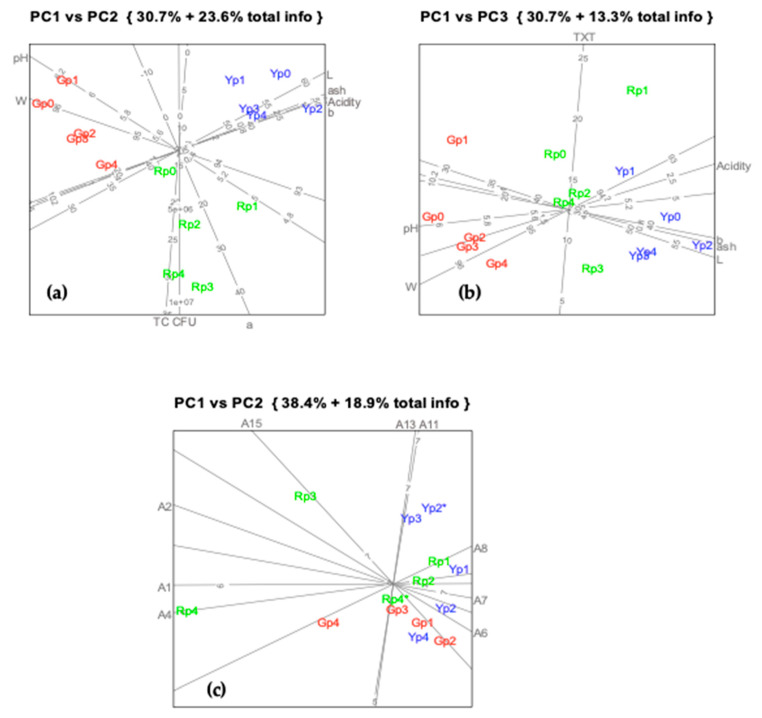
Predictive biplots of physicochemical, microbiological and sensory data over storage time applied to the plane PC1*vs*PC2 to evaluate quality changes over storage time: (**a**) PC1*vs*PC2 plot presenting most important variables and its relations (pH, moisture content (W), titratable acidity, ash content, microorganisms (CFU), total carotenoids (TC) and color parameters (*a*, b and *L*)); (**b**) PC1*vs*PC3 plot presenting important variables and its relations (pH, moisture content (W), titratable acidity, ash content, and color parameters *b* and *L* and texture (TXT)); (**c**) PC1*vs*PC2 plot presenting products attributes: surface moisture (A1), typical color (A2), brightness (A4), elasticity (A6), cohesiveness (A7), hardness (A8), firmness at first bite (A11), flavor typicity (A13), sourness (A15). Axis scales correspond to initial attributes values representing the scale used. Letters identify product type: Gp = green peppers, Rp = red peppers, Yp = yellow peppers associated to numbers (0 to 4) corresponding to sampling/storage time (5; 10; 14; 17 days, respectively). Axis scales correspond to initial variables values.

**Table 1 molecules-25-02323-t001:** Colorimetric parameters of bell peppers ^a^.

Bell Pepper	Days	*L^*^*	*a* ^*^	*b**	*C* ^b^	*h* ^c^
Gp	0	33.70 ± 0.03	−8.58 ± 0.12	16.9 3± 0.02	18.98 ± 0.12	116.87 ± 0.10
	5	33.60 ± 0.02	−7.78 ± 0.08	14.40 ± 0.01	16.37 ± 0.18	118.38 ± 0.07
	10	39.38 ± 0.03	−8.94 ± 0.04	18.65 ± 0.04	20.68 ± 0.03	115.59 ± 0.06
	14	33.83 ± 0.05	−7.81 ± 0.12	15.26 ± 0.08	17.15 ± 0.05	117.09 ± 0.02
	17	34.58 ± 0.04	−7.49 ± 0.05	14.95 ± 0.11	16.72 ± 0.02	116.63 ± 0.08
Rp	0	38.79 ± 0.02	31.37 ± 0.03	27.81 ± 0.03	41.92 ± 0.01	41.56 ± 0.01
	5	38.99 ± 0.08	33.73 ± 0.08	26.35 ± 0.06	42.80 ± 0.04	37.99 ± 0.02
	10	41.32 ± 0.07	30.02 ± 0.16	26.40 ± 0.04	39.98 ± 0.06	41.32 ± 0.06
	14	39.70 ± 0.08	30.89 ± 0.09	26.15 ± 0.13	40.48 ± 0.12	40.25 ± 0.04
	17	37.61 ± 0.04	28.79 ± 0.08	25.36 ± 0.11	38.37 ± 0.07	41.37 ± 0.03
Yp	0	61.39 ± 0.08	4.98 ± 0.09	49.98 ± 0.06	50.22 ± 0.15	84.31 ± 0.02
	5	54.51 ± 0.06	−0.09 ± 0.17	40.64 ± 0.07	40.64 ± 0.08	90.12 ± 0.06
	10	55.79 ± 0.03	−0.09 ± 0.12	46.47 ± 0.08	46.47 ± 0.04	90.10 ± 0.01
	14	52.47 ± 0.05	−0.21 ± 0.09	43.46 ± 0.04	43.46 ± 0.03	90.27 ± 0.04
	17	57.24 ± 0.07	−0.46 ± 0.04	41.82 ± 0.02	41.82 ± 0.02	90.62 ± 0.07

^a^ Data are expressed as the mean ± SD, n = 3; GP = Green Pepper; RP = Red Pepper; YP = Yellow Pepper; *L** = lightness value; *C*^b^ = Chroma value; *h*^c^ = hue angle expressed in degree.

**Table 2 molecules-25-02323-t002:** Shelf life of fresh bell peppers (*Capsicum annuum* L.) stored at different packaging conditions.

*Capsicum annuum* L.	Experiment Details:	Shelf-Life ^a^	Ref
Fresh-cut green bell peppers *cv Twingo* F1; Southern Greece	MAP; 5 and 10 °C, 90% RH Initial conditions-**P**: 22% O_2_; 2–5% CO_2_ Packaging: LDPE-60, MDPE-30 and PVC	14 days at 5 °C LDPE-60 film	[6]
Fresh-cut green bell peppers *cv Swarma* harvested from the Central Institute of Agricultural Engineering, Bhopal, India	Active Packages: **P1** (4.5% O_2_; 7.8% CO_2_); **P2** (4.7% O_2_; 7.5% CO_2_) 8 ± 1 °C, 95 ± 3% RH Packaging system designed by the authors using permeable polymeric films (38 μm thick polypropylene (PP) with ten perforations of 0.3 mm made using a needle).	49 days at 8 °C, 95% RH P1 and P2	[10]
Fresh-cut green bell peppers *cv Twingo* F1; Southern Greece	MAP; 0 and 5 °C, 90% RH **P1**: 5% O_2_; 10% CO_2_; **P2**: 5% O_2_; 15% CO_2_ Packaging: 20 μm thick HDPE (0.96g cm^−3^) packages	10 days 0 °C P1 and P2	[14]
Mature green bell peppers, obtained at R. Ranga Swamy farm (green house), Thanjavur, India	Initial conditions: 20% O_2_; 0.1% CO_2_ 33 °C, 65% RH; 20 °C, 75% RH; 8°C, 95% RH Packaging and storage conditions: Corrugated fiber board boxes; Diffusion channel system with different lengths (10, 1.5 and 25 cm)	10 days (33 °C/65% RH) 28 days (8 °C/95% RH) 25cm diffusion channel length	[24]
Mature green bell peppers, Orissa University of Agriculture and Technology, Bhubaneswar, Odisha, India	**P**: 9.0–9.8% O_2_; 3.3–4.1% CO_2_ 23–35 °C, 35–75% RH and 4–6 °C, 45% RH Packaging: MAP with LDPE (Low-density Polyethylene), MAP with PP, MAP in perforated LDPE films, MAP in perforated PP films-five holes of 0.3 mm diameter in each side of the film, shrink packaging with Biaxially Oriented PP (BOPP) film and vacuum packaging with PP film.	20 days 4–6 °C, 45% RH MAP with PP film and vacuum packaging with PP film	[25]
Fresh-cut red bell peppers var *Kannon* harvested from the Arava Desert of Israel	Initial conditions—**P**: 21% O_2_; 0.3% CO_2_ 1.5 °C; 7 °C and 17 °C, 94% RH Packaging: Xtend^®^ (XF) and Polyethylene (PE) films	14 days at 7 °C, 94% RH XF packaging	[26]
Mature green bell peppers; Cultivars–*Smooth Cayenne*^1^; *Sultan*^2^ Bangkerohan, Davao City, Philippines	Ambient Conditions: (28.74 ± 0.94 °C, 65.68 ± 7.43% RH; 13.62 KPa VPD) ^*^; Evaporative cooling (EC): (23.91 ± 3.85 °C, 93.84 ± 9.33% RH; 1.79 KPa VPD) ^**^ The EC was covered with two layers of jute sack (burlap) as walls of the EC cabinet. The sack wall was constantly bathed with water from a container placed on top of the EC cabinet to keep it moistened.	^1*^ 9 days ^1**^ 18 days ^2*^ 7 days ^2**^ 15 days	[27]
Fresh-cut green bell peppers; Jiangsu, China	Pressurized argon treatments (2–6 MPa/1 h) **P**: 5% O_2_; 8% CO_2_; 4 °C, 90% RH Packaging: ^Polystyrene^ packages; Pressurized Argon Treatments (PAT)	12 days (PAT of 4 MPa/1h); 4 °C	[28]
Red bell peppers in ripening stage, from green to red color, obtained at local distributor of Abidjan, Côte d’Ivoire	6, 16, 21 and 30 ± 2 °C; 20 days Packaging: Polyethylene perforated sealed bags (30 μm thick with 20 perforations of 5 mm in diameter.	12 days at 6 °C	[29]
Fresh-cut green bell peppers obtained at local distributor of São Paulo, Brazil (22°42′ S; 47°38′ W; 546 m)	**P**: 2% O_2_; 10% CO_2;_ 88% N_2;_ 1 ± 1 °C; 21 days Packaging: BOPP/LDPE plastic bags.	14 days at 1 ± 1 °C	[30]

**^a^** Best shelf life results due to the storage conditions proposed in the studies; **P**: Modified atmosphere packaging; ^1^*Smooth Cayenne*; ^2^*Sultan*; * Ambient Conditions: (28.74 ± 0.94 °C, 65.68 ± 7.43% RH; 13.62 KPa VPD; ** Evaporative cooling (EC): (23.91 ± 3.85 °C, 93.84 ± 9.33% RH; 1.79 KPa VPD).

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
