# Peer review of "Fresh-Cut Bell Peppers in Modified Atmosphere Packaging: Improving Shelf Life to Answer Food Security Concerns"

_molecules, 2020, doi:10.3390/molecules25102323_

Round 1

Reviewer 1 Report

Food safety measures have always been important in production and everyday life and have considerably increased in recent years because of microbial threat and pathogen infection of consumers.

In nutrition, multi-coloured bell peppers varieties are considered as an excellent source of vitamins (A, C), minerals (potassium, iron), and bioactive antioxidants. Appreciated by consumers, this vegetable is commercially important for many countries. In this perspective, the packaging methods focus on the preservation of the valuable nutrients in stored bell pepper especially with respect of microbial safety.

Authors of this article examined not only visual proprieties of the stored bell pepper but also performed e.g. moisture and pH measures and the antioxidant secondary metabolites content in each bell pepper variety. To better understand the permeability of the tested Krehalon MLF40 - PA/PE films they run appropriate analyses in modified atmosphere condition for 17 days at 5ºC. The results of PCA plots that combine multivariate factors seem to be quite promising for commercial purposes offering new improved storage conditions of bell pepper. To sum up I fully recommend publishing this article in Molecules journal.

I congratulate Authors well-written paper on multivariable analysis conducted on chemical and physical properties during storage of green, red and yellow bell peppers.

Few points could be improved:

Line 265 Description of lines in PCA plots lacks clarity and often are overlapped in the case of Fig. 6 (a) and (c)

In methods, simple titration with a KI solution for vitamin C content can be performed to complete the analysis of the various compounds in the manuscript. But of course, this can be done in future studies

Lines 645, 698, 711 Latin name of Capsicum annuum should be in italics

Please revise the order of citing authors, e.g. between [34] and [36] there is an gap

Author Response

Dear Editor and Reviewers,

            In attention to your kindly comments and suggestions for improving the quality in the Research Article entitled “Fresh-cut bell peppers into Modified Atmosphere Packaging: Improving shelf life to answer to food security concerns”, we highlight the changes we have made in the revised manuscript. English has been thoroughly revised and all insertions and changes in the manuscript are highlighted throughout the text. A point-by-point response is presented below.

Reviewer 1

(1) Line 265 Description of lines in PCA plots lacks clarity and often are overlapped in the case of Fig. 6 (a) and (c)

  • Following reviewer’s suggestion, names of variables that were overlapping in biplots of figures 6a and 6c are now separated and are easily readable. Also, figure capture was corrected.

(2) In methods, simple titration with a KI solution for vitamin C content can be performed to complete the analysis of the various compounds in the manuscript. But of course, this can be done in future studies

  • We appreciate the comment and inform you that further studies on the topic are underway.

(3) Lines 645, 698, 711 Latin name of Capsicum annuum should be in italics

  • All corrections were made.

(4) Please revise the order of citing authors, e.g. between [34] and [36] there is a gap.

  • The correction was made.

Reviewer 2 Report

In my opinion this is a very important and valuable work. Bioactive components were identified and detected with adequate methods. The results are very demonstrative.

Author Response

Dear Editor and Reviewers,

            In attention to your kindly comments and suggestions for improving the quality in the Research Article entitled “Fresh-cut bell peppers into Modified Atmosphere Packaging: Improving shelf life to answer to food security concerns”, we highlight the changes we have made in the revised manuscript. English has been thoroughly revised and all insertions and changes in the manuscript are highlighted throughout the text. A point-by-point response is presented below.

Reviewer 2

(1) In my opinion this is a very important and valuable work. Bioactive components were identified and detected with adequate methods. The results are very demonstrative.
(2) Special attention should be given to the following expressions.

• All changes and corrections suggested by reviewer 2 were made.

Line 16 „the” should be cancelled from the end of the line

Line 59 „is need” should be replaced by „are needed”

Line 118 „at” should be replaced by „on”

Line 119 „at” should be replaced by „on”

Line 176 „s” should be cancelled from the end of „phenols”

Line 208 „s” should be cancelled from the end of „phenols” and „anthocyanins”

Line 209 „s” should be cancelled from the end of „carotenoids”

Line 213 „s” should be cancelled from the end of „flavonoids”

Line 216 and 217 „what concerns” should be cancelled

Line 234 „s” should be cancelled from the end of „phenols, flavonoids, carotenoids”

Line 294 comma should be cancelled after „content”

Line 295 „representing” should be replaced by „represented”

Line 322 „in” should be replaced by „on”

Line 326 „s” should be cancelled from the end of „flavonoids”

Line 337 „neighbor” should be replaced by „neighbour”

Line 374 „are” should be inserted after „designs”

Line 462 „at” should be replaced by „on”

Line 464 „analyses” should be replaced by „analysis”

Line 466 „analyses” should be replaced by „analysis”

Line 506 „s” should be cancelled from the end of „phenolics”

Line 513 „s” should be cancelled from the end of „anthocyanins”

Line 514 „were” should be replaced by „was”

Line 516 „s” should be cancelled from the end of „contents”

Line 529 „were” should be replaced by „was”

Best regards,
Carla Barbosa
Thelma B. Machado
M. Rui Alves
M. Beatriz P.P. Oliveira